

# Multiphasic strain differentiation of atypical mycobacteria from elephant trunk wash

Kok-Gan Chan[1], Mun Fai Loke[2], Bee Lee Ong[3], Yan Ling Wong[2], Kar Wai Hong[1], Kian Hin Tan[1], Sargit Kaur[2], Hien Fuh Ng[2], MFA Abdul Razak[4] and Yun Fong Ngeow[2,5]

[1] Division of Genetics and Molecular Biology, Institute of Biological Sciences, Faculty of Science, University of Malaya, Kuala Lumpur, Malaysia
[2] Department of Medical Microbiology, Faculty of Medicine, University of Malaya, Kuala Lumpur, Malaysia
[3] Faculty of Veterinary Medicine, Universiti Malaysia Kelantan, Kelantan, Malaysia
[4] Department of Wildlife and National Parks, Peninsular Malaysia, Kuala Lumpur, Malaysia
[5] Faculty of Medicine and Health Sciences, Universiti Tunku Abdul Rahman, Kajang, Malaysia

Corresponding author
Yun Fong Ngeow,
yunngeow@yahoo.com

## ABSTRACT

**Background.** Two non-tuberculous mycobacterial strains, UM_3 and UM_11, were isolated from the trunk wash of captive elephants in Malaysia. As they appeared to be identical phenotypes, they were investigated further by conventional and whole genome sequence-based methods of strain differentiation.

**Methods.** Multiphasic investigations on the isolates included species identification with hsp65 PCR-sequencing, conventional biochemical tests, rapid biochemical profiling using API strips and the Biolog Phenotype Microarray analysis, protein profiling with liquid chromatography-mass spectrometry, repetitive sequence-based PCR typing and whole genome sequencing followed by phylogenomic analyses.

**Results.** The isolates were shown to be possibly novel slow-growing schotochromogens with highly similar biological and genotypic characteristics. Both strains have a genome size of 5.2 Mbp, G+C content of 68.8%, one rRNA operon and 52 tRNAs each. They qualified for classification into the same species with their average nucleotide identity of 99.98% and tetranucleotide correlation coefficient of 0.99999. At the subspecies level, both strains showed 98.8% band similarity in the Diversilab automated repetitive sequence-based PCR typing system, 96.2% similarity in protein profiles obtained by liquid chromatography mass spectrometry, and a genomic distance that is close to zero in the phylogenomic tree constructed with conserved orthologs. Detailed epidemiological tracking revealed that the elephants shared a common habitat eight years apart, thus, strengthening the possibility of a clonal relationship between the two strains.

## INTRODUCTION

Bacterial strains are genetic variants of a bacterial species or subspecies. Strain differentiation is usually carried out to establish clonal transmission in disease outbreaks, to confirm cross-infection in healthcare settings, or to study evolutionary diversity among bacteria. Traditionally, strains are differentiated within species by their biological properties, antigenic variations or susceptibility to bacteriophages. Since the advent of molecular biology, however, DNA-based methods have increasingly become the methods of choice for bacterial typing in modern laboratories. The more frequently used techniques include multilocus sequence typing (MLST), pulsed-field gel electrophoresis (PFGE), restriction fragment length polymorphism (RFLP) studies, and repetitive sequence-based PCR (rep-PCR) typing. These methods have been effectively used in the investigation of tuberculosis transmission and reactivation in human patients (*Gori et al., 2005*) and animals (*Aranaz et al., 2010*).

Unlike *Mycobacterium tuberculosis* which is an established pathogen in humans as well as a wide range of domestic, wild and captive animals (*Aranaz et al., 2010*), the non-tuberculous mycobacteria (NTM), also known as atypical mycobacteria, are mostly saprophytes in the natural environment, only occasionally implicated in disease. The investigation of a NTM disease often involves genotyping to establish a link with an environmental source of infection. Unfortunately, the large number of NTM species and subspecies and their variable typeability in different assays has made it difficult to find a common test for genotyping. The closest to a gold standard test appears to be PFGE which has been successfully used on both rapidly growing and slowly growing NTMs (*Jagielski et al., 2014*). The discriminatory power of newer methods has not been adequately studied. Nevertheless, RFLP analysis of insertion sequences such as IS *1245* and IS*1311* was found to be useful for the differentiation of *M. avium* subspecies isolated from animal and human sources (*Mijs et al., 2002*). Similarly, variable number tandem repeat (VNTR) typing methods have been applied successfully in *M. avium* and *M. avium* subsp. *paratuberculosis* (*Thibault et al., 2007*) and have been used to confirm clonal relationships between patient and environmental isolates of *M. ulcerans* (*Lavender et al., 2008*). MLST and randomly amplified polymorphic DNA PCR (RAPD) have facilitated investigations of NTM outbreaks in laboratories (*Zhang et al., 1997*; *Cooksey et al., 2008*).

Most of the animal pathogens reported in scientific literature are members of the *M. avium* complex, such as *M. avium* subsp. *paratuberculosis,* the cause of severe gastroenteritis in ruminants, *M. avium* subsp. *avium* in birds and raptors, and *M. avium* subsp. *hominissuis* in swine (*Aranaz et al., 2010*). In captive elephants, *M. avium* is also commonly isolated (*Payeur et al., 2002*) but is rarely associated with disease (*Yong et al., 2011*). The only two species reported to be the cause of elephant morbidity and mortality are *M. szulgai*, isolated from an acute fatal disease in two African elephants kept in a London zoo (*Lacasse et al., 2007*), and *M. elephantis* from the lung abscess of an elephant that died of a chronic respiratory illness (*Shojaei et al., 2000*). Generally, little is known about the epidemiology and zoonotic potential of NTMs among animals in captivity.

While studying tuberculosis among captive Asian elephants in Malaysia (*Ong et al., 2013*), we came across two strains of a slow-growing NTM that appeared to be identical in morphology and routine biochemical tests. We investigated the relationship between these two strains with conventional and whole genome sequence-based methods of strain differentiation. Our observations indicated two members of a possibly novel NTM species with highly similar genetic and protein profiles that characterize strains of clonal descent. Our whole genome sequence data also supported the reliability of conventional microbiological and molecular methods for the strain differentiation of non-tuberculous mycobacteria.

## MATERIALS AND METHODS

### Bacterial isolation

UM_3 and UM_11 were isolated from two Asian elephants (*Elephas maximus*) in 2012. UM_3 was from the offspring of a Malaysian jungle elephant. It was born in 1997 and was raised in a conservation center (KG) before its transfer to a zoo about 250 km away, in 2002. UM_11 was isolated from a 71-year-old elephant acquired from Assam, India, and brought to KG in 2010 under the aegis of the Department of Wildlife and National Parks, Peninsular Malaysia. Hence, both elephants had been in the same conservation center, eight years apart. Both had no overt signs of disease at the time of sampling. The collection of elephant trunk wash was approved by the Animal Care and Use Committee, Faculty of Veterinary Medicine, Universiti Putra Malaysia (reference number: UPM/FPV/PS/3.2.1.551/AUP-R163) (*Ong et al., 2013*).

Elephant trunk wash was collected using the "triple sample method", decontaminated with the modified Petroff method, and further processed as described previously (*Ong et al., 2013*). Mycobacterial isolation was carried out in the BACTEC Mycobacteria Growth Indicator Tube (MGIT) 960 System (Becton Dickinson, USA). Positive BACTEC cultures were sub-cultured onto Lowenstein-Jensen agar for incubation at 36 °C in light and in the dark, for up to 8 weeks, to observe growth rate and colonial morphology. The Tibilia test (Tibilia Genesis, Hangzhou, China) for the detection of *Mycobacterium tuberculosis*-specific MPB64 antigen was performed according to the manufacturer's instructions, to differentiate *M. tuberculosis* from non-tuberculous mycobacteria (NTM). For further investigations, isolates were subcultured on Middlebrook 7H10 agar with OADC supplement (BBL, Becton Dickinson).

### Molecular identification of isolates

DNA was extracted by boiling a few loopfuls of each isolate suspended in sterile distilled water, at 100 °C for 30 min, followed by centrifuging at 16,100 g for 2 min. The supernatant was used for a *hsp65*-based PCR (*Telenti et al., 1993*), using primers Tb11 (5′-ACC AAC GAT GGT GTG TCC AT) and Tb12 (5′-CTT GTC GAA CCG CAT ACC CT) to amplify a 441bp fragment in the gene. PCR products were purified using the QIAquick PCR Purification kit (QIAGEN, Germany) and sent for Sanger sequencing. The resultant DNA sequences were searched against the NCBI non-redundant nucleotide database using

BLAST web server (*Altschul et al., 1990*). The most probable species for each isolate was identified based on the nucleotide sequence similarity with reference strains.

## Biochemical tests

For conventional biochemical tests (*Kent, 1985*; *Babady & Wengenack, 2012*) and rapid biochemical profiling using API strips (bioMerieux, France), the inoculum used was a suspension of each isolate in normal saline, made up to McFarland 0.5 turbidity. For the Biolog Phenotype Microarray analysis, a turbidimeter was used to check the turbidity of the suspension and bacterial cells were added to achieve 81% T (transmittance). Microplates PM1, PM2, PM9 and PM10 were used with the OmniLog system (Biolog, USA). These microplates are 96-well microtiter plates containing different kinds of compounds. PM1 and PM2 test for carbon-utilization patterns while PM9 and PM10 test for tolerance to a wide range of osmolytes and pH (http://www.biolog.com/pmMicrobialCells.html). *M. tuberculosis* H37Ra was used as the control in all these tests.

## Protein profiling with liquid chromatography-mass spectrometry (Q-TOF *LC/MS*)

The NTM cells were lysed and proteins were extracted using the ProteoSpin detergent-free total protein isolation kit (Norgen Biotek, Canada) with the Halt protease and phosphatase inhibitors cocktail (Thermo Scientific, USA) included. The lysates were subsequently treated with 10 mM dithiothreitol (DTT; Bio-Rad, USA) at 37 °C for 10 min and alkylated with 55 mM iodoacetamide (IAA; Bio-Rad) for 30 min at room temperature. The proteins in the sample were digested with 1:50 (trypsin: protein) of MS-grade Pierce trypsin protease (Thermo Scientific) at 37 °C overnight. The samples were desalted using a Pierce C-18 spin column (Thermo Scientific) and dried to completeness in a refrigerated CentriVap centrifugal vacuum concentrator (Labconco, USA) before mass spectrometry analysis.

Tryptic peptides were analyzed on the 1260 Infinity HPLC-Chip/MS System (Agilent, USA). The HPLC-Chip was the Large Capacity C18 Chip (G4240-6210), which comprised a 160 nL enrichment column and a 150 mm analytical column. HPLC-grade water (0.1% formic acid) and acetonitrile (0.1% formic acid) were used as mobile phases A and B respectively. HPLC-grade acetonitrile and formic acid were procured from Friendemann Schmidt (Australia) and Sigma (USA), respectively. The Nanoflow gradient (%B) used was: 3% at 0 min, 3% at 5 min, 35% at 60 min, 40% at 67 min, 60% at 85 min, 60% at 95 min, 3% at 105 min; stop time: 120 min; nanopump flow rate: 0.3 uL/min; capillary pump gradient: 3% B isocratic; capillary pump flow: 2.5 uL/min; chip value position: enrichment at 95 min. An Agilent 6540 Accurate-Mass Q-TOF LC/MS System operated in positive ion mode was used for mass detection, applying the following settings: capillary voltage: 1850 V; drying gas flow: 5 L/min; drying gas temperature: 250 °C; fragmenter: 175 V; skimmer: 65 V; acquisition mode: autoMS/MS; scan range: 125–1,700 $m/z$ (MS), 50–1,700 $m/z$ (MS/MS); acquisition rate: 15 spectra/s (MS), 12 spectra/s (MS/MS); isolation width (MS/MS): narrow (∼1.3 $m/z$); collision energy: −4.8 (offset) + 3.6 (slope); max. precursors/ cycle: 15; active exclusion: enabled, exclude after one spectrum, release after 0.25 min; charge state preference: 2, 3 and >3, sorted

by abundance only; reference mass: 299.294457 and 1221.990637 *m/z* from continuous addition of trace amounts of methyl stearate and HP-1221 calibrant respectively into the ionization region. Reference mass correction was enabled. *De novo* sequencing was conducted with PEAKS Studio 7.0 (Bioinformatics Solutions Inc., Canada) with default parameters, except that: (i) parent mass error tolerance was 1.5 Da, (ii) fragment mass error tolerance was 0.5 Da, (iii) trypsin as digestion enzyme, (iv) carbamidomethylation (+57.02 Da, C) as fixed modification, (v) oxidation (+15.99 Da, M) as variable modification, (vi) maximum variable post-translation modification allowed per peptide was three and (vii) *Mycobacterium parascrofulaceum* ATCC BAA-614 UniProtKB reference proteomes database was used for identifications.

## Repetitive sequence-based PCR typing (Rep-PCR) with the diversilab system

Rep-PCR strain typing relies on the amplification of sequences between repetitive elements interspersed throughout the genome (*Healy et al., 2005*). For the typing of our strains, we used the automated Diversilab system from Spectral Genomics, Inc. (Houston, TX). Mycobacterial cells harvested from MGIT960 tubes were used for DNA extraction, using the Ultra Clean microbial DNA isolation kit (Mo Bio Laboratories, Inc, CA.), according to the manufacturer's instructions. The extracted DNA was evaluated in the Nanodrop ND-2000 spectrophotometer (Thermo Scientific, Wilmington, DE, USA) and diluted to 35 ng/μl for use in the DiversiLab system. The PCR was run on thermal cycler (Applied Biosystems, Foster City, CA) using the following conditions: initial denaturation at 95 °C for 5 min, followed by 35 cycles of denaturation at 95 °C for 30 s, annealing at 66 °C for 45 s, extension at 72 °C for 1 min, and final extension at 72 °C for 5 min. Each reaction constitutes 18.0 μl of rep-PCR MM1, 2.5 μl of GeneAmp 10 X PCR buffer, 2.0 μl of Primer Mix, 0.5 μl of AmpliTaq DNA Polymerase and 2 μl of template DNA. The detection of fluorescent rep-PCR fingerprint was carried out on micro-fluidic chips (LabChip device, Caliper technologies, Inc) on an Agilent model 2100 Bioanalyzer (Agilent Technologies, Inc, Palo Alto, CA). The subsequent fingerprint data was analyzed on the web-based DiversiLab v3.4.4 software (Bacterial Barcodes). The Top match option was used to find the closest match to the available DiversiLab *Mycobacterium* database in terms of similarity values based on the Pearson correlation coefficient. According to the manufacturer's guidelines, isolates were considered "different" if they had <95% similarity and ≥2 band differences for homogeneous organisms or ≥3 band differences for heterogeneous organisms; "similar" if there was <97% similarity and 1 band difference for homogeneous organisms or up to 2 band differences for heterogeneous organisms and "indistinguishable" if there was >95% similarity and no banding differences, including no variation in intensities of individual bands (*Shutt et al., 2005*).

## Whole genome sequencing and phylogenomic analyses

For each strain, a Nextera DNA sequencing library was prepared as described previously (*Ngeow et al., 2015*). Shotgun sequencing was performed with the Illumina HiSeq 2000 system. The sequencing reads generated were investigated using FastQC (fastqc)

(*Andrews, 2010*) and assembled with CLC Genomics Workbench v.7.0. Gene prediction was made using Prokaryotic Dynamic Programming Genefinding Algorithm (PRODI-GAL) Version 2.60 (*Hyatt et al., 2010*). The predicted CDSs were annotated by homology search against the NCBI nr database (ftp://ftp.ncbi.nlm.nih.gov/blast/db/). rRNA and tRNA were detected using RNAmmer 1.2 Server (*Lagesen et al., 2007*) and tRNAscan-SE 1.21 (*Schattner, Brooks & Lowe, 2005*). Phage elements were searched for using PHAST (*Zhou et al., 2011*).

The genomic data obtained was used to determine the degree of genetic relatedness between the two strains. For this purpose, we drew a Venn diagram with homologous genes identified as genes sharing >50% protein sequence identity, and constructed a Maximum Likelihood (ML) phylogenomic tree, using single copy orthologous CDSs generated from OrthoMCL (*Chen et al., 2006*), aligned with MUSCLE (*Edgar, 2004*) and reconstructed with 100 bootstrap replicates using Phyml 3.0 (*Guindon & Gascuel, 2003*). For comparison, 13 selected genomes were downloaded from the NCBI Genome database (http://www.ncbi.nlm.nih.gov/genome/?term=mycobacterium). They comprised (a) slow-growing schotochromogens *M. parascrofulaceum* ATCC BAA-614 (ADNV01), *M. yongonense* 05-1390 (CP003347), *M. tusciae* JS617 (AGJJ02), and *M. xenopi* (JAOC01), (b) a slow-growing photochromogen *M. kansasii* ATCC 12478 (NC022663), (c) non-pigmented slow-growers *M. avium* 104 (NC008595), *M. intracellulare* ATCC 13950 (NC016946), *M. colombiense* CECT 3035 (AFVW02), *M. genavense* ATCC 51234 (JAGZ01) and *M. smegmatis* str.MC2155 (NC_008596) (d) non-pigmented fast-growers *M. indicus pranii* MTCC 9506 (NC018612) and *M. septicum* DSM 44393 (CBMJ02), and (e) a rapid-growing schotochromogenic strain *M. iranicum* UM_TJL (AUWT01). *Nocardia farcinica* (NC_006361) was used as the out-group.

## ANI and TETRA analyses

For the calculation of average nucleotide identity (ANI) values (*Nei & Li, 1979*), we determined all conserved genes in the two genomes by whole-genome sequence comparisons using the BLAST algorithm. Genes were considered conserved when they had a BLAST match of at least 30% overall sequence identity and at least 70% identity in the aligned region. Generally, strains sharing more than 94% ANI are considered to belong to the same species. For the analysis of tetra-nucleotide usage patterns, we used TETRA (http://www.megx.net/tetra) (*Teeling et al., 2004*) to calculate the correlation coefficients of the tetra-nucleotide signatures in the two genomes. Correlation coefficient values above 0.99 suggest a high probability of two strains being from the same species.

## Multilocus Sequence (MLS) analysis

We compared the allelic variants in seven housekeeping genes (*argh, cya, gnd, murC, purH, rpob* and *pta*) in UM_3 and UM_11. Full-length gene sequences retrieved from the two annotated genomes were concatenated and aligned with MAFFT software (http://mafft.cbrc.jp/alignment/server/). We compared fragment sizes of the seven loci and looked for polymorphic sites and SNPs.

## RESULTS

### Morphology, biochemistry and PCR-sequencing

Both UM_3 and UM_11 grew after 10–14 days of incubation at 36 °C on LJ agar. The colonies were small, bright yellow, schotochromogenic, discrete and moist-looking (Figs. S1A and S1B). They did not grow at 45 °C or in 5% w/v sodium chloride but grew at 22 °C and in the presence of streptomycin 2 mg/L (Table S1). The bacilli were acid-fast with the Ziehl-Neelson stain and were negative for the MPB64 antigen in the Tibilia test for *M. tuberculosis*. PCR-sequencing of the *hsp65* gene identified both strains as *M. paracrofulaceum* with 99% coverage and 96% sequence similarity.

Standard biochemical tests (Table S1) showed both strains to be negative in iron uptake, niacin production, urea hydrolysis and oxidase tests, but strongly positive in the 68 °C catalase and semi-quantitative catalase tests. Of a total of 87 tests in three API kits (API 20NE, API Coryne and API 50 CH), only seven (8.0%) gave different results: the fermentation/assimilation of glucose, maltose, lactose, mannitol, *N*-acetyl-glucosamine, AMD starch and potassium 5-Keto-gluconate G. In the Biolog assays, of 384 substrates surveyed, 24 (6.3%) differentiated the two strains. These differences mostly involved the utilization of organic acids as carbon sources and the ability to grow in 3 to 6% sodium salts (data not shown).

### Protein profiles

The LC/MS data yielded 495,933 MS scans and 156,008 MS/MS scans. After filtration, 4,089 peptide-spectrum matches were obtained and a total of 1,206 proteins were identified with 0.6% FDR. Among the proteins identified, the majority (1160/1206 or 96.2%) were highly conserved in both strains; only 46 were differentially expressed. Using the PEAKS label-free quantification tool, 5 proteins had more than 2-fold higher expression in UM_3 compared to UM_11 (Fig. 1). These proteins (Elongation factor Tu [tr|D5PC89|]), 60 kDa chaperonin [tr|D5PBK4| ], DNA-binding protein HU (tr|D5PIY7|], Cyclic nucleotide-binding domain protein [tr|D5P8V4|] and transcription elongation factor Gre [tr|D5PFQ0|]) seem to be involved in transcription and translation regulation. In addition, 22 metabolic proteins (14 were expressed in UM_3 and 8 were expressed in UM_11), 5 ribosomal proteins (UM_3) and 2 bacterial secretion proteins (UM_11) were also uniquely expressed among these two strains by sequence homology search tool (SPIDER) of the PEAKS software (Fig. S2). However, without further functional analysis, we were unable to show correlation between the expression of these proteins and our API and selected Biolog phenotyping test results.

### Rep-PCR typing

Our rep-PCR typing results showed 98.8% similarity between the two strains in both the dendrogram (Fig. 2) and the overlay (Fig. 3). By the manufacturer's guidelines, the two strains should be classified as almost indistinguishable, as they showed an identical banding pattern with differences seen only in the intensities of a few individual bands (Figs. S3A and S3B).
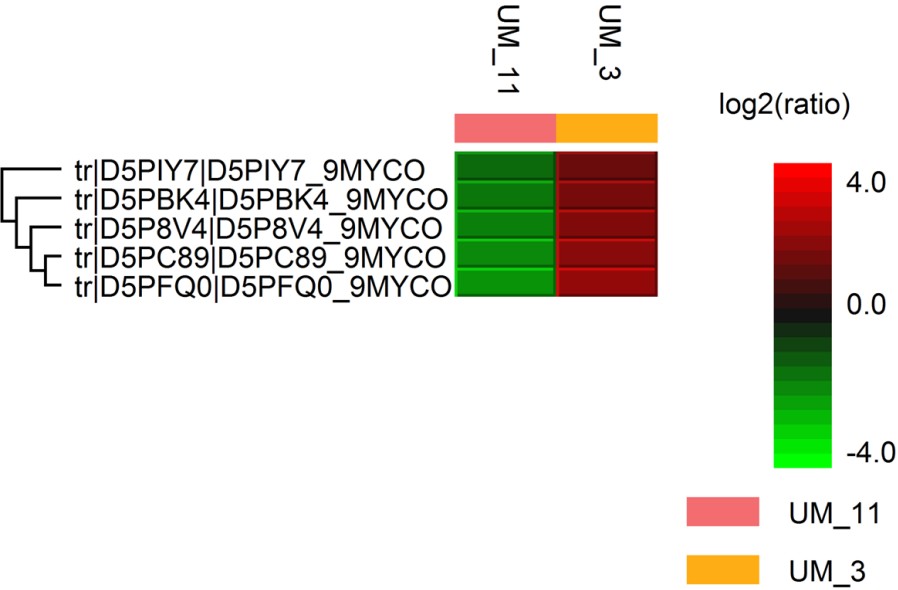

**Figure 1  Hit map of differential protein profile of UM_3 and UM_11 generated by PEAKS Studio 7.0.** Tr|D5PIY7|: DNA-binding protein HU; tr|D5PBK4|: 60 kDa chaperonin; tr|D5P8V4|: cyclic nucleotide-binding domain protein; tr|D5PC89|: elongation factor Tu and tr|D5PFQ0|: transcription elongation factor Gre.

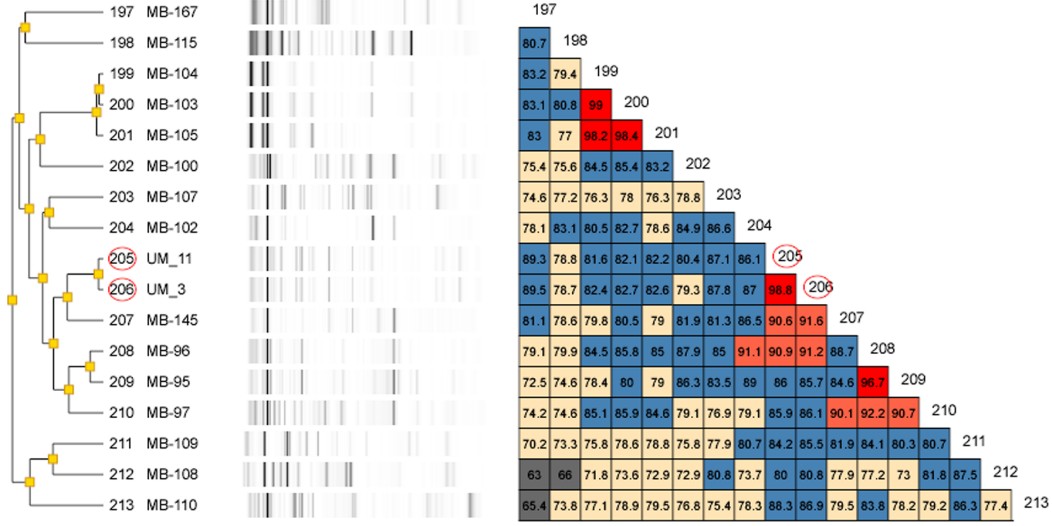

**Figure 2  Dendrogram of UM_3 and UM_11 with other Mycobacterium isolates in the DiversiLab Mycobacterium database.** UM_3 and UM_11 are 98.8% similar.

## Genomic analyses

The Illumina sequencing generated 5,236,492 reads assembled into 177 contigs for UM_3 (at an average coverage of 72 times and N50 of 98,626bp) and 6,279,982 reads assembled into 275 contigs for UM_11 (at an average coverage of 59 times and N50 of 126,029bp). The number of putative coding sequences (CDS) predicted was 4958 for UM_3 and 5022

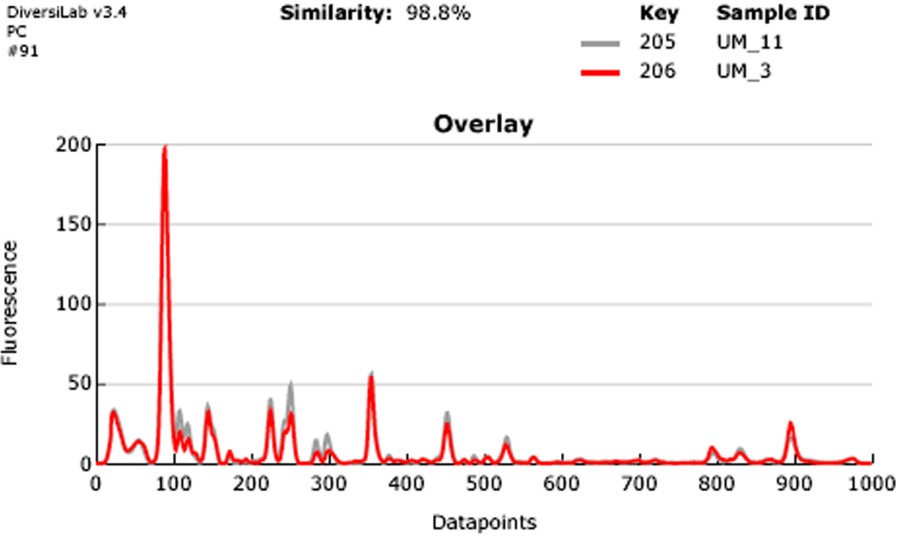

**Figure 3 Overlay of rep-PCR fingerprints of UM_3 and UM_11.**

for UM_11. Both draft genomes contained 5.2Mbp with a G+ C composition of 68.8%.
By the best BLAST hit results, 68.7% −70.21% of their coding sequences mapped towards
*M. parascrofulaceum*. Each genome had one rRNA operon (5S, 16S and 23S), 52 tRNAs and
two phage-related elements which were the phage T7 F exclusion suppressor FxsA and a
phage-related replication protein.

The ANI values showed 99.98% nucleotide similarity between the two strains compared
to 88.43% similarity between each strain and *M. parascrofulaceum*. The TETRA correlation
coefficient was 0.99999 between both strains and 0.99666 (UM_3) to 0.99678 (UM_11)
between the two strains and *M. parascrofulaceum*, suggesting that both strains belong to
the same species that is distinct from *M. parasrofulaceum* (Table S2).

In the orthologs-based phylogenomic tree, both UM_3 and UM_11 were clustered
together next to *M. parascrofulaceum,* with a genomic distance of 0.0000 base substitutions
between them (Fig. 4). The Venn diagram (Fig. 5) showed the two strains sharing 4,752 of
4,797 (99.1%) gene families in the core genome (Table S3). In the MLS sequence analysis
of seven housekeeping genes, all corresponding genes in both strains were of the same
gene length and had identical allele sequences except for the murC gene which showed two
possible indels (2bp and 97 bp) in the 5′ end of the gene (Fig. 6).

## DISCUSSION

As NTM species are ubiquitous in natural environments, isolates from elephant trunk wash
are usually considered environmental saprophytes colonizing the trunk or contaminating
trunk wash during sample collection. It is conceivable, however, that respiratory infections
can occur via the inhalation of aerosols from contaminated feed and water sources, and
cross-infection can occur among elephants, during trumpeting and trunk-spraying. In
epidemiological investigations, the route of transmission is established by demonstrating
the presence of the infecting strain in the suspected source (whether environmental or

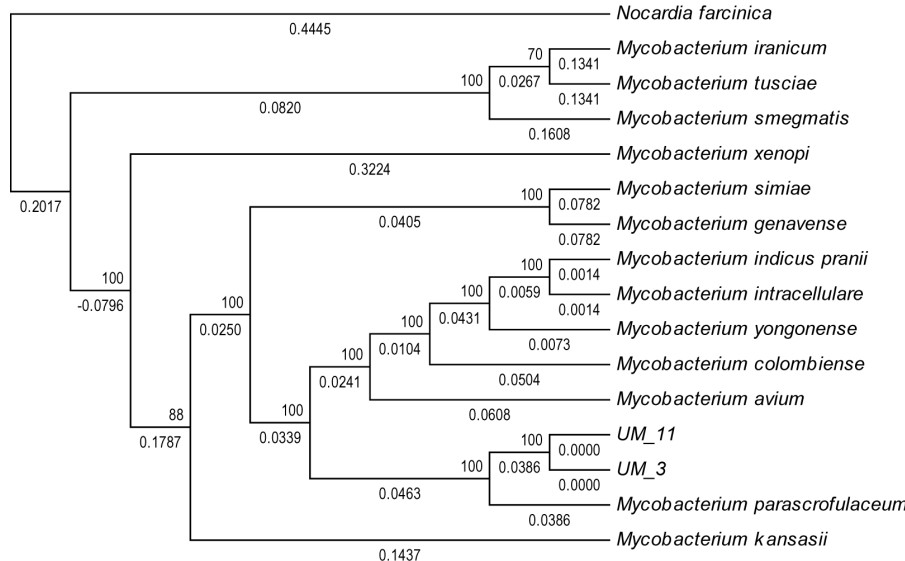

**Figure 4 Orthologs-based phylogenetic tree showing UM_3 and UM_11 in relation to other mycobacterial spp.** Nocardia farcinica NC0063 is the out-group.

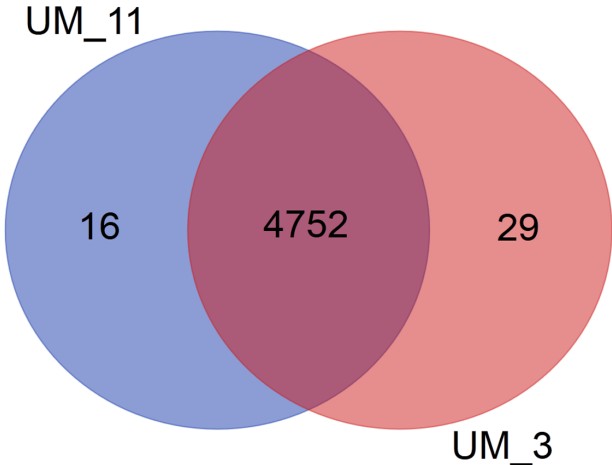

**Figure 5 Venn diagram showing sharing of gene families between UM_3 and UM_11.**

animal) and in the affected animals. The two NTM strains in our study shared multiple biological features that indicated a high probability that they were strains of clonal descent from a common source. To study this possibility, we attempted strain differentiation using a multiphasic approach, as conventional bacterial typing methods have not been adequately evaluated for most NTM species.

In addition to routine biochemistry, we demonstrated phenotypic similarity with metabolic and protein profiling. The Biolog Phenotype MicroArray system that allows bacterial profiling based on their ability to grow and respire on compounds serving as carbon, nitrogen, phosphorus or sulfur sources, has been shown to be able to distinguish both slowly growing mycobacteria such as *M. tuberculosis* and *M. leprae,* as well as rapidly

```
UM_3_murC      --gtgtataagagaca-----------------------------------------
UM_11_murC     atgtgtataagagacaggttcgagctggtgggcaacgccgggtcggtgcgcgtctacgac
               **************

UM_3_murC      ---------------------------------------------------ggtcctc
UM_11_murC     gactacgctcaccatcccaccgagatcagcgccacgctggccgcggtgcgcacggtcctc
                                                                       *******

UM_3_murC      gagcagagcggcggcggtcgcagcgtggtggtgtttcagccccacttgtattcgcgcaca
UM_11_murC     gagcagagcggcggcggtcgcagcgtggtggtgtttcagccccacttgtattcgcgcaca
               ************************************************************

UM_3_murC      aaggaattcgcggcagagttcggccgcgcgctcgatgccgccgacgaggtgttcgtcctc
UM_11_murC     aaggaattcgcggcagagttcggccgcgcgctcgatgccgccgacgaggtgttcgtcctc
               ************************************************************

UM_3_murC      gacgtctacggcgcgcgtgagcaaccgctcgccggcgtcagcggcgccagcgtcgccgag
UM_11_murC     gacgtctacggcgcgcgtgagcaaccgctcgccggcgtcagcggcgccagcgtcgccgag
               ************************************************************

UM_3_murC      cacgtcagcgtgccggtgcgctacctgccgaacttctccgcggtccccgagcaggtggcg
UM_11_murC     cacgtcagcgtgccggtgcgctacctgccgaacttctccgcggtccccgagcaggtggcg
               ************************************************************

UM_3_murC      gccgcggccgggcccggggacgtcatcgtcacgatgggcgcgggtgacgtgaccctgctg
UM_11_murC     gccgcggccgggcccggggacgtcatcgtcacgatgggcgcgggtgacgtgaccctgctg
               ************************************************************

UM_3_murC      ggcccggagatcgtgaccgagctgcggatccgggacaaccgcagcgcgcccggccggccg
UM_11_murC     ggcccggagatcgtgaccgagctgcggatccgggacaaccgcagcgcgcccggccggccg
               ************************************************************

UM_3_murC      ggggcgccgcggtga
UM_11_murC     ggggcgccgcggtga
               ***************
```

**Figure 6  Alignment of murC gene sequences in UM_3 (396bp) and UM_11 (495 bp).**

growing NTMs such as *M. smegmatis, M. fortuitum, M. chelonae* and *M. phlei* (*Rahman, Jaques III & Daniels, 2008*). For UM_3 and UM_11, the Biolog tests showed only a minor difference in energy source utilization and salt tolerance. The comparable metabolic profiles of the two strains correlated well with their highly conserved protein profiles obtained with LCMS, suggesting that they were functionally very similar. However, the protein profiling could have been limited by the lack of a reference proteome for our strains. We had to use *M. parascrofulaceum* ATCC BAA-614 as our reference, a strain that showed the highest *hsp65* sequence identity but only around 70% coding gene sequence similarity with UM_3 and UM_11.

For genotyping, we used the Diversilab system which has been adapted for use on mycobacteria. It was reported to show either equivalent or higher discriminatory power than that of IS6110-RFLP for the typing of *M. tuberculosis* (*Jang et al., 2011*). Among the NTM, it has been used effectively for the strain differentiation of the *M. avium* complex (*Cangelosi et al., 2004*) as well as *M. abscessus* isolates (*Zelazny et al., 2009*). Going by the interpretation guidelines recommended by the manufacturer, UM_3 and UM_11 which showed more than 98% band pattern similarity, were almost indistinguishable and could very well be descendants of the same clone.

More comprehensive information for strain differentiation came from the comparison of whole genome sequences that showed both strains to be identical in their genome size, G + C content, number and sequence of tRNAs, number and sequence of phage elements, ANI values and tetranucleotide signatures, as well as 99% conservation in gene families. The larger number of unique genes in UM_3, as shown in the Venn diagram, suggested that it might be the progenitor to UM_11 which appeared to have lost some genes. This observation ties in well with the history of UM_3 being in the conservation center years before UM_11 was brought there.

We compared the sequences of seven housekeeping genes for both strains as MLS has been shown to be not only reliable for outbreak investigations but also potentially useful for strain identification and species designation of novel isolates (*Bielaszewska et al., 2011*). In the usual application of MLS typing (MLST) for a bacterial species, each unique combination of locus variants is assigned a sequence type (ST), which is compared to STs in a public MLST database for that bacterial species. In our case, since our strains did not have a species designation, we could not assign a definite ST to each strain; instead, we compared the allele profiles generated by concatenating the seven gene sequences together for the comparison of sequence length and variation. The almost identical profiles obtained for UM_3 and UM_11 are consistent with the high genetic similarity demonstrated by the rep-PCR typing and the assumption of clonal descent.

Unlike bacterial species differentiation that is guided by differences in 16S RNA sequences and percentage DNA-DNA hybridization, there is no consensus guideline for bacterial strain differentiation. In genotyping, molecular markers are used to generate distinct DNA fingerprints. These markers are conserved alleles that have remained stable over much of evolution (*Owen & Xerry, 2003*). Unfortunately, the discriminatory power of commonly-used molecular markers has not been adequately studied for most NTMs, especially highly conserved species with members sharing very similar genetic fingerprints. Furthermore, some widely-prevalent genotypes can become predominant in an environment. This situation can give rise to incorrect interpretation of clonality among strains belonging to the same genotype. For these reasons, in an epidemiological investigation, strains showing similar DNA fingerprints cannot be interpreted to be clonally-related unless an epidemiological link is established as well.

In our study, the history of a shared habitat for our two elephant hosts allowed us to assume a clonal descent for the almost indistinguishable UM_3 and UM_11. The minor differences in their gene expression could be due to their individual responses to environmental stimuli, and their minor genetic differences could be the result of genetic drift in the course of their continuous microevolution (*Israel et al., 2001*). Overall, our findings showed consistency in results from different types of analysis and supported the reliability of current biological and molecular strain differentiation methods for NTMs.

## CONCLUSION

Strain differentiation with multiple approaches can potentially unveil important phenotypic and genotypic features worthy of further investigations. In our two closely

related NTM strains, we showed differentially expressed proteins that could be involved in the regulation of transcription and translation as well as other metabolic and secretory activities. Downstream investigations might lead to the elucidation of gene regulation mechanisms that affect the appearance, survival and behavior of mycobacterial spp.

Our use of whole genome sequencing demonstrated obvious advantages of this approach over conventional genotyping. From the whole genome backbone, multiple nucleotide and gene sequences can be extracted and used in diverse ways to provide information for taxonomic classification, strain differentiation and the inference of evolutionary relationships. Nevertheless, more genetic studies on NTMs from different sources, geographic areas and times would help the development of more reliable genotyping methods applicable to all NTM species, without the need for species-specific modifications.

### Funding
This work was supported by research grants UM.C/625/1/HIR/MOHE/CHAN/14/4 (H50001-00-A000038), UM.C/625/1/HIR/MOHE/CHAN/01 (A-000001-50001), UM.C/625/1/HIR/MOHE/CHAN/14/1 (H-50001-A000027) and UM.C/625/1/HIR/MoE/CHAN/13/3 (H-50001-A000030) from the University of Malaya, Kuala Lumpur, Malaysia. The funders had no role in study design, data collection and analysis, decision to publish, or preparation of the manuscript.

### Grant Disclosures
The following grant information was disclosed by the authors:
University of Malaya, Kuala Lumpur, Malaysia: UM.C/625/1/HIR/MOHE/CHAN/14/4 (H50001-00-A000038), UM.C/625/1/HIR/MOHE/CHAN/01 (A-000001-50001), UM.C/625/1/HIR/MOHE/CHAN/14/1 (H-50001-A000027), UM.C/625/1/HIR/MoE/CHAN/13/3 (H-50001-A000030).

### Competing Interests
The authors declare there are no competing interests.

### Author Contributions
- Kok-Gan Chan contributed reagents/materials/analysis tools, reviewed drafts of the paper.
- Mun Fai Loke contributed reagents/materials/analysis tools, prepared figures and/or tables, reviewed drafts of the paper.
- Bee Lee Ong, Sargit Kaur and MFA Abdul Razak performed the experiments, reviewed drafts of the paper.
- Yan Ling Wong, Kar Wai Hong and Kian Hin Tan performed the experiments, analyzed the data, prepared figures and/or tables, reviewed drafts of the paper.
- Hien Fuh Ng performed the experiments, prepared figures and/or tables, reviewed drafts of the paper.

- Yun Fong Ngeow conceived and designed the experiments, analyzed the data, wrote the paper, reviewed drafts of the paper.

## Animal Ethics

The following information was supplied relating to ethical approvals (i.e., approving body and any reference numbers):

The collection of elephant trunk wash was approved by the Animal Care and Use Committee, Faculty of Veterinary Medicine, Universiti Putra Malaysia (reference number: UPM/FPV/PS/3.2.1.551/AUP-R163) (*Ong et al., 2013*).

## DNA Deposition

The following information was supplied regarding the deposition of DNA sequences:

The whole-genome sequences have been deposited in GenBank: LATB00000000 (Mycobacterium sp. UM_3) and LATC00000000 (Mycobacterium sp. UM_11).

## Supplemental Information

Supplemental information for this article can be found online at http://dx.doi.org/10.7717/peerj.1367#supplemental-information.

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
