# Peer review of "Multiphasic strain differentiation of atypical mycobacteria from elephant trunk wash"

_PeerJ, doi:10.7717/peerj.1367_

## Round 0.1 · original submission · Major Revisions

The current version of the manuscript is not scientifically sound. There are several scientific errors in the manuscript as pointed by the Reviewers. Furthermore, the Reviewer 3 has raised some ethical concerns. Kindly justify them.

·

Basic reporting

no comment

Experimental design

no comment

Validity of the findings

no comment

Additional comments

The author should recheck the format how to write the pattern of journal in all of references. For example, the name of journal line 393, Journal of Clinical Microbiology, but the same journal in line 411, Journal of clinical microbiology. Please recheck all of the references.

Reviewer 2 ·

Basic reporting

The introduction part not very exciting as the abstract and the results in the manuscript. Author should incorporate some specific changes in order to enhance the possible readership. As the manuscript’s main finding is the characterization and differentiation of two NTM strains by genotyping and whole genome analysis, I would prefer the introduction section should focus mainly on the molecular techniques used for NTM strain differentiation and should also mention how these techniques could be useful in NTM strain differentiation (infectious and non-infectious) by citing specific reference. They should then expand the introduction by mentioning how genotypic studies on NTM could significantly improve our understanding about the epidemiology and zoonotic potential of NTM. I think this should be the main focus in the introduction section.

At the end of the introduction part, author should provide
1) a comprehensive description of the objective set for the work,
2) specific results obtained
3) one line-concise and clear interpretation of the results obtained

The conclusion part is shallow in meaning

Experimental design

Experimental design is fine

Validity of the findings

Results
Morphology, biochemistry and PCR-sequencing
This part is not been fully represented as there is no evidence of the study has been presented in the manuscript, either in main text or in supplementary.
For example:
1) Photographs of morphological characterization, and colony formation
2) A Table with recorded reading from biochemical and metabolic profiling expt
3) Growth curve related to growth of strain at 45°C, in 5% w/v sodium chloride, and at 22°C and in the presence of streptomycin 2 mg/L.
Rep-PCR typing
The gel picture of rep-PCR on gDNA (at least 2-3 independent extraction) from two strains should be included in the supplementary section of the manuscript.

Additional comments

Major comments
The manuscript of Chan et al. describes the polyphasic approach for differentiation of nontuberculous mycobacterial (NTM) strains from elephant trunk wash. NTM represents primary and opportunistic pathogen to animals as well as zoonotic agents. Therefore, in-depth genotypic studies, such as those based whole genome-sequence, on NTM is critically important as the spectrum of disease caused by them is constantly increasing. These studies are also important for ascertaining the epidemiology and zoonotic potential of NTM. The manuscript is a work of merits. The content of the manuscript is good as well as the structure is also good. I would suggest major revisions in the manuscript.
Among the points of strength of the work, there are the following:
1) The abstract is concise and has been written nicely
2) The materials and methods are described nicely
3) Result- although fine but there are some comments that need to be corrected/included in the manuscript
Among the points of weakness of the work, there are the following:
1) The introduction part not very exciting as the abstract and the results in the manuscript. Author should incorporate some specific changes in order to enhance the possible readership. As the manuscript’s main finding is the characterization and differentiation of two NTM strains by genotyping and whole genome analysis, I would prefer the introduction section should focus mainly on the molecular techniques used for NTM strain differentiation and should also mention how these techniques could be useful in NTM strain differentiation (infectious and non-infectious) by citing specific reference. They should then expand the introduction by mentioning how genotypic studies on NTM could significantly improve our understanding about the epidemiology and zoonotic potential of NTM. I think this should be the main focus in the introduction section.
2) A number of useful techniques such as Biolog Phenotype Microarray analysis, protein profiling with liquid chromatography-mass spectrometry (Q-TOF LC/MS) and automated rep-PCR genotyping have been used in this study; however, results obtained from these studies need to be interpreted better- detailed below.
3) The conclusion part is too shallow- detailed below.

Minor section-wise suggestions in order to improve the work are as below:
Abstract
Abtract is fine
Introduction
Page 4 Line 54: Reference in the text need to be formatted as Aranaz et al., 2010 and in the other parts of the manuscript.
Page 5 Line 75: While studying tuberculosis……………… strain differentiation. In this paragraph author should: 1) provide comprehensive description of the objective set for the work, and 2) provide specific results obtained from morphological (there are specific results), and also from the conventional as well as whole genome based strategies that were used for differentiation of two NTM strains.

Materials and methods
Page 6 Line 105: Molecular identification of isolates
Authors have used hsp65-based PCR RFLP, and therefore, it is necessary to mention about the restriction profiling. Authors mentioned that “PCR products were purified using the QIAquick PCR Purification kit (QIAGEN, Germany) and sent for Sanger sequencing”. However, there is no description regarding RFLP. There is reference related reference cited but a single line mentioning about RFLP in Molecular identification of isolates is required. If PCR-RFLP profiles was done In-silico using mapdraw (dnastar) or similar, please specify it.
Page 7 line 114: Biochemical tests
Author should rephrase following sentence as meaning is not clear.
“A fresh culture of each NTM strain was suspended in normal saline to McFarland 0.5 turbidity for conventional biochemical tests…..”
Please also specify why McFarland turbidity std no. 0.5 was used- for ex. for antimicrobial susceptibility testing or other
Page 8 Line 137: Protein profiling with liquid chromatography-mass spectrometry (Q-TOF LC/MS)
Author should rewrite the sentence as follow
HPLC-grade acetonitrile and formic acid were respectively procured from Friendemann Schmidt (Australia) and Sigma (USA). There is no need to put information regarding the MilliQ.
Page 9 Line 158: Repetitive sequence-based PCR typing (Rep-PCR) with the Diversilab System
The information regarding primers used for rep-PCR is missing
Page 10 Line 177: I would prefer ‘diverse’ instead of ‘different’

Results
Morphology, biochemistry and PCR-sequencing
This part is not been fully represented as there is no evidence of the study has been presented in the manuscript, either in main text or in supplementary.
For example:
1) Photographs of morphological characterization, and colony formation
2) A Table with recorded reading from biochemical and metabolic profiling expt
3) Growth curve related to growth of strain at 45°C, in 5% w/v sodium chloride, and at 22°C and in the presence of streptomycin 2 mg/L.
Protein profiling
Page 13 Line 255: Author should provide implication of the result obtained from protein profiling. Do the five proteins (that were differentially expressed in the studied strains) have any relevance with other growth, metabolic, phenotypic and functional studies conducted? In fact, the heat map (Page 22 Fig 1) indicating that the five proteins are upregulated in UM_3 strain but the functional relevance of upregulation of these five proteins in UM_3 is missing.

Rep-PCR typing
The gel picture of rep-PCR of gDNA (at least 2-3 independent extraction) from two strains should be included in the supplementary section of the manuscript.

Discussion
The discussion part is satisfactory
Conclusion
The conclusion part is little disappointing- though, the statements made are relevant but are very shallow in meaning and do not render findings exciting or worthy to note. However, there are some interesting findings, especially related with differential protein expression in two strains as well as differential gene expression results as detailed in Supplementary table 2. I would suggest authors to conclude manuscript showing functional implication of data obtained from protein and gene expression studies.

·

Basic reporting

The quantum of work is appreciable, but the fundamental flaw in the study is with the objective of the study. The author do not mention anything why only two animals were subjected for this investigation, if they were asymptomatic. How many other elephants and/or other animal species are hosted in the conservation center. The another concern this reviewer has regarding the circumstances under which an Indian elephant was taken to Malaysia? If this was legal transport, or part of research collaboration, the same should have been mentioned. This reviewer considers it an unethical research, if these details are not declared by the authors.

Experimental design

Molecular work is standard, but whether it is automation in analytical laboratory or even high tech molecular method including WGS, the general principle of "garbage in garbage out" stands fast. The authors have not provided details of aseptic techniques practiced during the sample collection and processing. This limitation is further accumulated in the study design, when the authors are silent on the constituents of the culture medium. It is well known that for isolating MAP and M. bovis special growth supplements are required. From the manuscript it is not known if these were added or not?. If these supplements are not added , in that case one can convincingly presume that naturally occurring/infecting species in these animals were missed and only contaminants were grown ( from soil or skin or the devices used for sample collection). It is not difficult to understand that if the main organism is missed in the primary culture, subsequent cultures or molecular method will not be able to pick the missed pathogen. It is further suspected, because the culture was presumably (author roles are not given) done in medical laboratory, which might have used the conventional MGIT culture medium made for isolating human mycobacteria.

Validity of the findings

Validity of findings that these isolates were single pathogen/ saprophytes isolated from these animals is doubtful. Though sequence details are not doubted. Ethics and methods of primary isolation are major issues.

Additional comments

Please see comments above. If the authors can address these questions, major revisions will be required.

---

## Round 0.2 · accepted · Accept

The manuscript is suitable for publication.

Reviewer 2 ·

Basic reporting

Authors have satisfied all the raised comments to improve the quality of the manuscript. I think now the manuscript is in acceptable form.

Experimental design

Satisfied

Validity of the findings

Satisfied